# Impact of Stepwise Recruitment Maneuvers on Cerebral Hemodynamics: Experimental Study in Neonatal Model

**DOI:** 10.3390/jpm13081184

**Published:** 2023-07-25

**Authors:** Teresa Torre Oñate, Antonio Romero Berrocal, Federico Bilotta, Rafael Badenes, Martin Santos Gonzalez, Laura de Reina Perez, Javier Garcia Fernandez

**Affiliations:** 1Department of Anaesthesiology, Intensive Care and Pain, Hospital Universitario Puerta de Hierro en Majadahonda, 28222 Majadahonda, Spain; antonromero@hotmail.com (A.R.B.); ventilacionanestesia@gmail.com (J.G.F.); 2Department of Anaesthesiology and Intensive Care, Sapienza University of Rome, 00185 Rome, Italy; bilotta@tiscali.it; 3Department of Anaesthesiology, Intensive Care and Pain, Hospital Clinic Universitari en Valencia, University of Valencia, 46010 Valencia, Spain; rafaelbadenes@gmail.com; 4Medical and Surgical Research Unit, Instituto de Investigación Sanitaria Puerta de Hierro-Segovia de Arana, Hospital Universitario Puerta de Hierro en Majadahonda, 28222 Majadahonda, Spain; martin.santos@salud.madrid.org; 5Department of Neurosurgery, Hospital Universitario Puerta de Hierro en Majadahonda, 28222 Majadahonda, Spain; lauradereina@hotmail.es

**Keywords:** recruitment maneuvers, neonate, cerebral hemodynamics, intracranial pressure, cerebral oximetry

## Abstract

Background: Lung recruitment maneuvers (LRMs) have been demonstrated to be effective in avoiding atelectasis during general anesthesia in the pediatric population. Performing these maneuvers is safe at the systemic hemodynamic and respiratory levels. Aims: We aimed to evaluate the impact of a stepwise LRM and individualized positive end-expiratory pressure (PEEP) on cerebral hemodynamics in an experimental neonatal model. Methods: Eleven newborn pigs (less than 72 h old, 2.56 ± 0.18 kg in weight) were included in the study. The LRM was performed under pressure-controlled ventilation with a constant driving pressure (15 cmH_2_O) in a stepwise increasing PEEP model. The target peak inspiratory pressure (PIP) was 30 cmH_2_O and the PEEP was 15 cmH_2_O. The following hemodynamic variables were monitored using the PICCO^®^ system: mean arterial pressure (MAP), central venous pressure (CVP), and cardiac output (CO). The cerebral hemodynamics variables monitored were intracranial pressure (ICP) (with an intraparenchymal Camino^®^ catheter) and cerebral oxygen saturation (rSO_2_) (with the oximetry monitor INVOS 5100^®^ system). The following respiratory parameters were monitored: oxygen saturation, fraction of inspired oxygen, partial pressure of oxygen, end-tidal carbon dioxide pressure, Pmean, PEEP, static compliance (Cstat), and dynamic compliance (Cdyn). Results: All LRMs were safely performed as scheduled without any interruptions. Systemic hemodynamic stability was maintained during the lung recruitment maneuver. No changes in ICP occurred. We observed an improvement in rSO_2_ after the maneuver (+5.8%). Conclusions: Stepwise LRMs are a safe tool to avoid atelectasis. We did not observe an impairment in cerebral hemodynamics but an improvement in cerebral oxygenation.

## 1. Introduction

Mechanical ventilation is a lifesaving approach that carries inherent risks including brain damage and cognitive impairment [1,2,3]. The use of lung recruitment maneuvers (LRMs) is widely applied in many different settings (thoracic, general, and cardiac surgery but also in critically ill patients) since these types of maneuvers provide a protective mechanism against damage associated with mechanical ventilation by reducing the driving pressure (DP) and making the distribution of the tidal volume (TV) more homogeneous [4,5,6]. However, there are other areas, such as neurosurgery and the neurocritical patient, where the use of positive end-expiratory pressure (PEEP) has been classically related to increased intracranial pressure (ICP), which is secondary to decreased cerebral venous return due to increased intrathoracic pressure. In these cases, the application of LRMs and PEEP are very controversial or even contraindicated depending on the patient’s situation [7,8,9]. Nevertheless, there are studies that show how ventilating neurocritical patients with PEEP is a safe procedure [10,11,12]. In addition, the data obtained from adults are different from those obtained from the pediatric population [13,14]. The effects in patients with an alteration to the blood–brain barrier are not the same as in patients without acute brain injury.

Experimental studies have demonstrated the efficacy of LRMs in achieving optimal alveolar opening during mechanical ventilation in neonatal models.

Evidence supports the safety of employing recruitment maneuvers in neonatal models [15]. LRMs have been found to have no deleterious effects on systemic hemodynamics (including arterial blood pressure, central venous pressure, and cardiac output), as well as on respiratory parameters (air trapping and pneumothorax) [16,17]. However, the impact of these maneuvers on cerebral perfusion in neonatal brain models remains unknown.

This study was designed to evaluate the impact of a stepwise LRM and individualized PEEP on cerebral hemodynamics (ICP and cerebral oxygen saturation) in an experimental neonatal model.

## 2. Materials and Methods

### 2.1. Ethics Committee

This study has been approved by the Institutional Animal Care and Use Committee of the Puerta de Hierro-Segovia Arana Health Research Institute, Madrid, Spain (Ethical Committee CEA 008/2017 Ref PROEX 234/17). This study was conducted in accordance with the ARRIVE guidelines for experimentation in animal research (Animal Research: Reporting of In Vivo Experiments). The animals were handled according to the European and national regulations for the protection of experimental animals (2010/63/UE and RD 53/2013). This study included Landrace–Large White newborn pigs, less than 72 h old and 2.56 ± 0.18 kg in weight. All were examined by a veterinarian prior to experimentation.

### 2.2. Anesthesia

Anesthesia was delivered using the approach previously described [15]. General anesthesia induction was pursued with 8% sevoflurane at 2.5 L/min with the lowest FiO_2_ possible to maintain peripheral O_2_ saturation (SpO_2_) ≥ 92%. A 24G peripheral line was cannulated. A bolus of morphic chloride (0.2 mg/kg) and a bolus of cisatracurium (0.15 mg/kg) were administered. Once an adequate anesthetic depth was reached (2.5% sevoflurane exhaled fraction, which is the newborn swine MAC) [18], the piglet was intubated with an uncuffed 3.5 mm diameter endotracheal tube (ETT). To guarantee the absence of leaks during the LRM, we opted to secure the ETT to the trachea. Surgical dissection was performed in planes, exposing the trachea. Once exposed, a tracheal tube was placed and sealed with a ribbon wrapped around the trachea to prevent any leaks. This sealing with a ribbon wrapped around the trachea also prevents any injury or occlusion of the trachea. We confirmed that the ligature effectively sealed the ETT without causing any narrowing of its diameter, based on the absence of any observable changes in pressures before and after the trachea was ligated. The absence of leaks was determined through the analysis of the flow volume.

The animals were connected to a Flow-i C20 anesthesia machine (Getinge, Solna, Sweden) under volume-controlled ventilation (VCV) for a TV of 6 mL/kg and a respiratory rate (RR) that maintained normocapnia: an end tidal CO_2_ (EtCO_2_) between 40 and 45 mmHg, without generating auto PEEP (30–40 rpm during basal ventilation and an RR of 35 rpm during the LRM). Basal PEEP was set to 3 cmH_2_O, and an I:E ratio of 1:1. The starting FiO_2_ was 30%. A convective warming device (Equator^®^) was used to maintain the piglet in normothermia (37–39 °C) with measurement of central temperature taken with an esophageal thermometer. Anesthesia maintenance was accomplished with sevoflurane at 2.5%, morphine chloride boluses at 0.2 mg/kg/h, and a continuous perfusion of cisatracurium at 0.12 mg/kg/h. Maintenance fluid therapy was maintained with Ringer’s lactate at 4 mL/kg/hour.

No volume loading or other drugs were routinely administered prior to the experiments. According to the protocol, we proceeded as follows:-10 mL/kg of physiological saline at 36 °C was administered when the mean arterial pressure (MAP) was under 45 mmHg.-An extra dose of morphine (0.1 mg/kg) was administered when the piglet had a heart rate greater than 180 bpm.

### 2.3. Monitoring

Electrocardiography and pulse oximetry were monitored in all animals with the Infinity Delta monitor, Dräger^®^. The right jugular vein was cannulated with a Seldiflex^®^ (67714J18, Promimed, Cedex, France) 5.5F and 8 cm long three-lumen central venous catheter. This vascular access was used for the administration of fluids and medication, as well as for central venous pressure (CVP) monitoring and thermodilution.

For invasive hemodynamic monitoring, the right femoral artery was cannulated with a PULSIOCATH 3F arterial thermodilution catheter, 7 cm in length (PV2013L07-A) from the PiCCO monitor^®^ (PULSION Medical Systems AG, Munich, Germany).

ICP monitoring was achieved with the placement of an ICP monitoring catheter Camino^®^ Model 1104B. This catheter was placed by performing a trephine at the right frontal level with a drill nº 36 (2.71 mm) in prone position. The pressure transducer was placed through a compressor plug, leaving it at the intraparenchymal level.

For regional rSO_2_ monitoring, we used the INVOS 5100^®^ system (Somanetics Corporation, Troy, MI, USA) by placing a unilateral neonatal sensor at the left frontal level because ICP monitoring at the right frontal level prevented bilateral sensor placement.

At the end of all monitoring techniques, we left the newborn pigs in the supine position.

### 2.4. Recruitment Maneuvers

The LRMs were run using the Flow-i 4.3 anesthesia system’s lung recruitment-specific software^®^ (Figure 1). Parameters were set as follows: pressure-controlled ventilation with a DP of 15 cmH_2_O and a baseline PEEP of 3 cmH_2_O in a stepwise increasing PEEP model (increment at each step: 5 cmH_2_O, first step: 5 cmH_2_O). Other settings: RR 35 rpm, target peak pressure: 30 cmH_2_O, and target PEEP: 15 cmH_2_O. The I:E ratio during the LRM was 1:1. Five breaths were set at each pressure step and ten breaths at the maximum pressure step. Pressure reductions were made in the VCV mode with a TV of 6 mL/kg. We descended from a PEEP of 8 cmH_2_O with PEEP decreases of 2 cmH_2_O, providing 30 s in each step to calculate the alveolar collapse point (first step in which Cdyn drops) and individualized PEEP (value of PEEP prior to the collapse point). Subsequently, the opening maneuver was repeated (up to a peak pressure of 30 cmH_2_O and PEEP of 15 cmH_2_O) to end the LRM with the calculated individualized PEEP (according to the best Cdyn value). LRMs should be stopped at any moment if desaturation (SpO_2_ < 91%) or hypotension occur (MAP < 30 mmHg or <20% of baseline).

### 2.5. Recorded Variables

Respiratory monitoring: inspiratory peak pressure, mean airway pressure (Pmean), inspiratory and expiratory TV (ICV and ECV), PEEP, dynamic compliance (Cdyn), static compliance (Cstat), EtCO_2_, FiO_2_, and SpO_2_.

Arterial blood gases were drawn prior to each LRM (basal ventilation), and 10 min after the end of the LRM, pO_2_ and pCO_2_ were analyzed.

Systemic hemodynamic monitoring: heart rate (HR); blood pressure (BP): SBP (systolic blood pressure), DBP (diastolic blood pressure), and MAP; CVP; and cardiac output (CO) measurements were obtained using thermodilution and arterial pulse wave analysis.

Cerebral hemodynamic monitoring: CPP was calculated with the formula (CPP = MAP − ICP), ICP, and rSO_2_.

Every parameter was continuously monitored. The aforementioned data were compared at these times: at the start of the maneuver, during the maximum opening pressure step, at each step of PEEP descent, and 10 min after the end of the LRM.

We decided to evaluate the baseline data (start of the LRM) following the induction of general anesthesia and intubation, as these procedures may contribute to partial lung collapse. The selection of maximum opening pressure steps allowed us to evaluate the effects of increased intrathoracic pressure on systemic hemodynamics and cerebral parameters. It also allowed us to examine the response and potential adverse effects of the maneuver at the maximum pressure level. The last selected time point data comparison was 10 min after completing the LRM, aiming to capture values when alveoli are maximally expanded. This specific temporal point allowed us to evaluate the immediate effects of the LRM and any associated changes in lung function, systemic hemodynamics, and cerebral parameters.

### 2.6. Statistical Analysis

Sample size and power calculations were based on previous data from similar studies and determined using the GRANMO 7.12 software program (Institut Municipal d’Investigació Mèdica, Barcelona, Spain). Eleven animals per group were deemed adequate to accept an alpha risk of 0.05 and a beta risk of 0.05 in a two-sided test to obtain a statistically significant difference > 1 unit in Cdyn and a standard deviation assumed to be 0.9.

A descriptive analysis was performed for categorical variables using absolute and relative frequencies, and for numerical variables using the median, 25th and 75th percentiles, and minimum and maximum values. To evaluate the evolution over time of the different parameters, generalized estimating equation (GEE) models were used, which consider that each animal was evaluated at different points in time. The “identity” function was used as the link function and a Gaussian distribution was used for the dependent variables. The dependent variables were ICP, rSO_2_, MAP, CPP, CVP, CO, and EtCO_2_. The time variable was introduced as a fixed independent variable, and the marginal effects were obtained for each time with their respective 95% confidence intervals, as well as a graphical representation of them. To evaluate the effectiveness of the LRM, a comparison was made between parameters before and after the maneuver (Cdyn, Cstat, PaO_2_, Pmean, and PEEP), using the nonparametric Wilcoxon signed-ranks test. The significance level was set to 0.05 for all comparisons. The statistical package used is Stata/IC v.15.1 (StataCorp. 2017. Stata Statistical Software: Release 15. College Station, TX, USA: StataCorp LLC).

## 3. Results

A total of 12 newborn pigs were studied, but one of them was excluded due to hemodynamic collapse after induction of anesthesia. The LRM was performed according to protocol in all experimental animals and a complete set of data was recorded. No additional boluses of saline or morphine chloride were necessary.

### 3.1. Effects on Respiratory Variables

Comparing the respiratory data obtained at the baseline with those obtained after the application of individualized PEEP shows that improvements were noted in all measured parameters, including PaO_2_, Cdyn, and Cstat. However, it is worth noting that only the parameter Cstat demonstrated statistical significance (*p* = 0.042).

During the LRM we observed a higher individualized PEEP compared to the initial programmed PEEP: PEEP 3 (3;3) vs. PEEP 6 (5;6) (*p* = 0.038) (Table 1). Analyzing the evolution of EtCO_2_ throughout the LRM shows that there were no significant changes (Figure 2).

### 3.2. Effects on Systemic Hemodynamics

The mean baseline MAP was 53.2 mmHg (±6.8), the mean baseline CVP was 6.2 mmHg (±2.29), and the mean baseline CO was 1 l/min (±0.24). All three systemic hemodynamic parameters remained constant throughout the different phases of the LRM (Table 2).

### 3.3. Effects on Cerebral Hemodynamics

The baseline CPP was 44 mmHg (±7.43). It remained constant throughout the different phases of the LRM (Table 2). The baseline ICP was 9.2 mmHg (2.25). The maximum ICP value was in Aperture 2 at 11.2 (95% CI 8.55; 13.84) but without statistically significant differences (*p* = 0.226). The lowest ICP value was with PEEP 3 at 9.1 (95% CI 7.23; 10.96), which was also not statistically significant (*p* = 0.941) (Figure 3).

Another variable analyzed was rSO_2_ with a baseline value of 49% (±4.34). No statistically significant variations were observed in the moments of maximum alveolar pressure, but a statistically significant improvement was observed at the end of the LRM; the level at the individualized PEEP was +5.8% over baseline rSO_2_ (95% CI 0.28; 11.31) (*p* = 0.04) (Figure 3).

#### 3.3.1. Analysis of the Relationship between ICP and Changes in Hemodynamic Variables

The variations in MAP did not have a statistically significant influence (*p* = 0.640) on the behavior of ICP throughout the LRM. Instead, a relationship between CPP and ICP was observed in our study. When CPP increased by 1 mmHg, ICP decreased by 0.09 mmHg (95% CI −0.16; −0.28) (*p* = 0.006). Furthermore, a relationship between CVP and ICP was also observed, with an increase in ICP of 0.92 mmHg (95% CI 0.6; 1.23) (*p* < 0.001) for each mmHg of CVP. CO had no statistically significant effect (*p* = 0.399) on ICP during the LRM (Figure 4).

#### 3.3.2. Analysis of the Relationship between ICP and Respiratory Parameters

On analyzing ICP, we observed that, with respect to Pmean, there was a discrete increase in ICP as Pmean increased; however, these changes were not statistically significant (*p* = 0.118). By contrast, EtCO_2_ turned out to have an impact on ICP significantly, since for every 1 mmHg increase in exhaled CO_2_, ICP increased by 0.2 mmHg (95% CI 0.14; 0.28) (*p* < 0.001) (Figure 4).

#### 3.3.3. Analysis of the Relationship between rSO_2_ and Hemodynamic Parameters

In relation to MAP, we observed that for each mmHg increase, rSO_2_ increased by 0.39% (95% CI 0.29; 0.5) (*p* < 0.001). With respect to CPP, rSO_2_ increased by 0.35% (95% CI 0.24; 0.46) (*p* < 0.001) for each mmHg that CPP increased. In contrast to ICP, there was an effect of CO on rSO_2_, since for each l/min increase in CO, rSO_2_ improved by 7.22% (95% CI 1.64; 12.80) (*p* = 0.011). CVP also influenced the behavior of rSO_2_, since for each mmHg increase in CVP, rSO_2_ also increased by 1.2% (95% CI 0.63–1.77) (*p* < 0.001) (Figure 5).

#### 3.3.4. Analysis of the Relationship between rSO_2_ and Respiratory Parameters

Changes along the LRM of rSO_2_ as a function of Pmean did not show statistical significance (*p* = 0.119). We observed instead an increase in rSO_2_ in relation to the increase in EtCO_2_; for each mmHg that EtCO_2_ increased, rSO_2_ also increased by 0.21% (95% CI 0.07; 0.34) (*p* = 0.03) (Figure 5).

## 4. Discussion

In this experimental study, we evaluated the impact of LRMs on ventilation and systemic and cerebral hemodynamics. The recorded data confirm, with the protocol used in the present study, the efficacy of LRMs in achieving optimal alveolar opening with an improved Cstat. The data also confirm that LRMs do not impact the ventilation and systemic hemodynamics as previously reported [15]. The present experimental series demonstrates that there are no changes in ICP, there is an improvement in rSO_2_, and CPP remained stable throughout the entire LRM sequence.

Our model is a neonatal piglet with intact brain function. We chose an experimental model because invasive neurological and hemodynamic monitoring in healthy neonates cannot be performed for ethical reasons. However, we believe it is very important to have data and scientific evidence on the behavior of the brain in neonatal patients where alterations in ICP and cerebral hemodynamics can have devastating consequences on neurological development.

The implications of an LRM and PEEP on ICP can be explained with the Monro–Kellie theory. ICP tends to remain constant up to certain limits. ICP depends on the volume of three components (the brain, vascular compartment, and cerebrospinal fluid). When the volume of one of these compartments increases, the volume of the other two is reduced as a compensatory mechanism to maintain a constant ICP. There is a point in the ICP/intracranial volume ratio at which these protective mechanisms cease to be effective and ICP increases exponentially with low volume increases [19]. In ventilated patients, this situation occurs earlier when they have decreased cerebral compliance [20].

Although the generalized knowledge is that LRMs and PEEP alter ICP, there are other studies that support our findings. Feldman studied in an experimental model how PEEP affected rabbits with an intracranial mass. He did not find an increase in ICP, but he found that these compensatory mechanisms were less effective, and the point of decompensation was reached earlier. In our case, we did not observe an increase in ICP because our patients have an intact central nervous system with intact protective mechanisms [21].

Therefore, we can say that an LRM with an opening pressure between 15 and 30 mmHg during fewer than 25 breaths (43 s), and in a stepwise manner with individualized PEEP, did not seem to have cerebral repercussions.

Regarding hemodynamic effects, neither MAP nor CPP decreased in our study. CO remained constant throughout the entire LRM, which justifies the hemodynamic stability at BP levels. This lack of impact on CO can be explained by the fact that the newborn pigs were normovolemic (baseline CVP of 6.2 mmHg), which would explain why no serum boluses were required during the LRM.

Sevoflurane was the anesthetic agent chosen for our study. This inhalational anesthetic has been the subject of numerous studies investigating its impact on systemic hemodynamics. Our decision was based on its ability to maintain adequate hemodynamic stability with minimal effects on blood pressure and systemic resistance. Additionally, sevoflurane has demonstrated the ability to maintain stable cardiac outputs during anesthesia, making it a safe choice [22].

Regarding its relationship with intracranial pressure (ICP) and regional cerebral oxygen saturation (rSO_2_), several studies have examined these aspects in patients undergoing intracranial surgery. Sevoflurane has been found to exert neuroprotective effects, as it can reduce ICP and improve rSO_2_ compared to other anesthetic agents. Furthermore, studies have shown that sevoflurane decreases the cerebral metabolic rate of oxygen (CMRO_2_) while maintaining a proportional reduction in cerebral blood flow (CBF) [22,23]. This preserved ratio of CBF to CMRO_2_ supports its utility in neuroanesthesia, indicating its potential to preserve cerebral perfusion and mitigate neuronal damage.

Studies in different populations with brain pathology support our observations. Frost observed that, in adults with brain damage, ICP did not increase even with PEEP levels of 40 cmH_2_O in the absence of falling CO or BP [10]. Pulitanò, who studied children with brain tumors, demonstrated that increasing the PEEP improved pulmonary compliance without producing an increase in ICP or a decrease in MAP or CPP. They also observed an increase in CVP with PEEP compared to ventilation without PEEP [14].

Caricato et al. studied the effects of pressurization (PEEP 0 to 12 cmH_2_O) in patients with altered brain function (traumatic brain injury or subarachnoid hemorrhage). No significant changes in ICP or cerebral compliance were observed in any case, but they determined an increase in CVP and a decrease in MAP and CPP in patients with normal pulmonary compliance [24]. The observed differences when compared with our results may be due to the difference in the patients’ blood volume status.

It should be noted that CPP depends on ICP but also on BP. Our newborn pigs maintained adequate CPP for two reasons: optimal levels of ICP and optimal levels of MAP. Videtta found, similarly to previous studies, that with increasing intrathoracic pressure, CPP remained stable despite increasing ICP (possibly because the increase in ICP was minimal) [25,26]. Other studies found neither reductions in MAP or CPP, nor increases in ICP [13].

In most of the studies, an increase in CVP was observed. We observed that CVP directly increased proportionally to the Pmean. The rise in intrathoracic pressure favors the increase in pressure in the right atrium and in turn, retrogradely, in the right jugular vein. This generates a decrease in venous drainage of the sagittal sinus, thus increasing ICP [27].

There are protective mechanisms to attenuate the increase in ICP: distensibility and the presence of valves in the jugular veins, and drainage in the vertebral venous plexus. An increase of 20 mmHg in the right atrium is necessary to overcome the protection generated from collapsed jugular veins [28].

In our study, there were no changes in CVP and ICP. This is possibly because our LRMs have a very limited duration in time (less than 43 s). In addition, most studies do not consider either Pmean or DP [27,29].

Regarding lung–brain interactions, there is a relationship between ICP and CO_2_. The increase in arterial CO_2_ pressure generates vasodilation that conditions a rise in cerebral perfusion with increased ICP and increased jugular venous saturation. Mascia and Robba found that increased PEEP only had an effect on ICP in patients who did not respond to alveolar recruitment. An excessive increase in PEEP produces hyperinflation with alveolar overdistension, leading to an increase in dead space [30,31,32,33].

Our final objective was to determine if LRMs improved cerebral oxygenation despite the hemodynamic changes it could produce. In our study, rSO_2_ improves by almost 6% at the step where individualized PEEP is established. There are few studies in the literature on the effects of protective ventilation on cerebral hemodynamics. Nemer observed an improvement in SpO_2_ and in the cerebral tissue pressure of O_2_ in patients with TBI (with an associated ARDS) by increasing PEEP to 15 cmH_2_O without ICP elevation [34].

Other studies aimed to assess the relationship between PaCO_2_ and PEEP with rSO_2_ and optic nerve sheath diameter. There is a direct relationship between this diameter and ICP and it is a non-invasive way of assessing it. It was observed that rSO_2_ increased with PEEP. The optic nerve sheath diameter increased as well (also indicating an increase in ICP), but it never exceeded physiological values. Hyperventilation of the patients improved ICP but decreased rSO_2_, as we observed [31,35].

The main limitations of our study are that it was performed in a neonatal animal model and therefore cannot be directly extrapolated to humans, although the study of newborn pigs is the most commonly used for neonatal studies and the one that most closely resembles humans.

Another consideration is that our patients have an integrated blood–brain barrier, so we cannot extrapolate the results to patients with brain damage. Regarding the number of newborn pigs, eleven may seem a small sample size, but the reason we used this size is because of ethical constraints. It is important to include the minimum number of animals to obtain significant results. Additionally, in experimental animal research, the usual number per group to obtain significant results is between six and ten in most of the studies.

There is scarce evidence in the literature on the behavior of ICP during RMs in healthy newborns. We decided to develop this study in an experimental animal model since it is not ethically justified to perform invasive procedures in healthy neonates. We thought it would be interesting to evaluate rSO_2_ since it is a tool that provides great informative value with few complications associated with its use. The results obtained indicate that these maneuvers enhance cerebral oxygen supply and are regarded as safe interventions from a hemodynamic, respiratory, and cerebral perspective in neonatal patients without neurological impairments. Consequently, they can be employed in different clinical settings (operating rooms or intensive care units). In the future it would be important to study these effects in damaged model brains as well.

## 5. Conclusions

In our study, we have observed that LRMs in a healthy neonatal model are safe and well tolerated hemodynamically at the systemic and cerebral levels. The LRM does not produce a significant increase in ICP but does improve final cerebral oxygenation. We consider that it is important to confirm that the patient is in a state of euvolemia before performing an RM to avoid hemodynamic alterations. Clinical studies in humans are needed to verify these findings. As our study uses an experimental animal design, it is limited to a small sample size due to the ethical limitations of the ARRIVE guidelines. However, we calculated the sample size to be sufficient to obtain statistically significant results. It would be interesting to conduct clinical studies with an increased number of patients to verify whether our results are confirmed in studies with larger sample sizes.

## Figures and Tables

**Figure 1 jpm-13-01184-f001:**
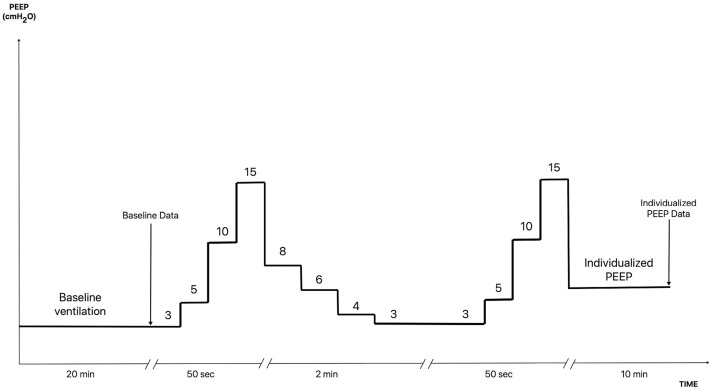
Schematic of the RM used in this study. Constant driving pressure (15 cmH_2_O). Increasing PEEP (5 cmH_2_O). Rising branch with pressure-controlled ventilation and falling branch with volume-controlled ventilation. Baseline values were recorded before the RM (after 20 min of protective lung ventilation with a PEEP of 3 cmH_2_O) and individualized PEEP values were recorded 10 min after performing the RM.

**Figure 2 jpm-13-01184-f002:**
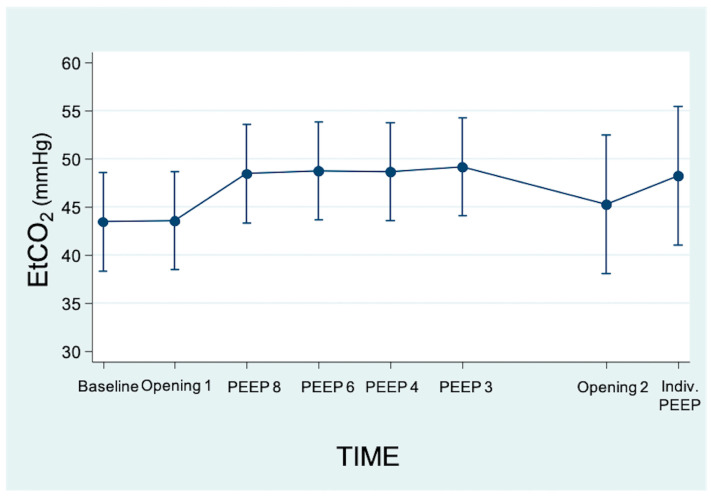
Linear prediction of changes in EtCO_2_ (end-tidal carbon dioxide pressure) throughout the recruitment maneuver (95% CI). There were no statistically significant changes with respect to the baseline step.

**Figure 3 jpm-13-01184-f003:**
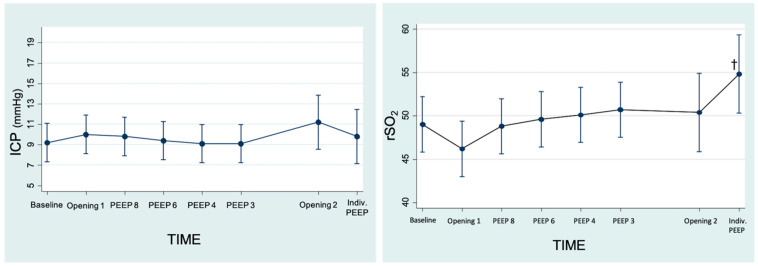
Linear prediction of changes in ICP (intracranial pressure) and changes in rSO_2_ (regional cerebral O_2_ saturation) throughout the recruitment maneuver (95% CI). There were no statistically significant changes with respect to the baseline step in ICP. † There was an improvement in rSO_2_: *p* = 0.04.

**Figure 4 jpm-13-01184-f004:**
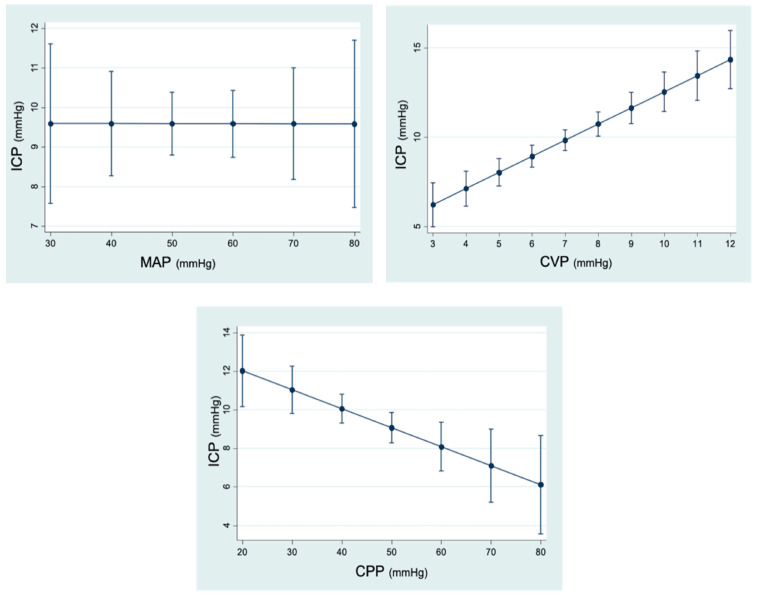
Linear prediction of ICP (intracranial pressure) according to hemodynamic and respiratory variables (95% CI). MAP (mean arterial pressure): *p* = 0.640. CVP (central venous pressure): *p* < 0.001. CPP (cerebral perfusion pressure): *p* = 0.006. CO (cardiac output): *p* = 0.399. EtCO_2_ (end-expiratory CO_2_): *p* < 0.001.

**Figure 5 jpm-13-01184-f005:**
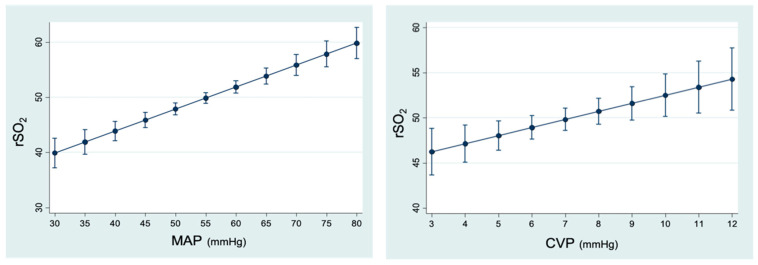
Linear prediction of rSO_2_ (regional cerebral O_2_ saturation) according to hemodynamic and respiratory variables (95% CI). MAP (mean arterial pressure): *p* < 0.001. CVP (central venous pressure): *p* < 0.001. CPP (cerebral perfusion pressure): *p* < 0.001. CO (cardiac output): *p* = 0.011. EtCO_2_ (end-expiratory CO_2_): *p* = 0.003.

**Table 1 jpm-13-01184-t001:** Comparison of the respiratory variables (median and its interquartile range 25–75) assessed before and after the lung recruitment maneuver using the Wilcoxon signed-ranks test.

	Baseline Situation	Individualized PEEP	*p* Value
PaO_2_	115 (94.7–128)	122 (119–149)	0.34
Cdyn	3 (2–3)	4 (4–4)	0.09
Cstat	4 (2.9–4.1)	4.3 (3.9–4.7)	0.04
Pmean	7.5 (7.5–8)	8.5 (8–9)	0.17
PEEP	3 (3–3)	6 (5–6)	0.04

Abbreviations: PaO_2_, arterial O_2_ partial pressure (mmHg); Cdyn, dynamic compliance (mL/cmH_2_O); Cstat, static compliance (mL/cmH_2_O); Pmean, mean airway pressure (cmH_2_O); PEEP, peak end-expiratory pressure (cmH_2_O).

**Table 2 jpm-13-01184-t002:** Hemodynamic variables analyzed throughout the lung recruitment maneuver. Linear prediction using generalized estimation equation.

	Opening 1	*p* Value	Opening 2	*p* Value	Individualized PEEP	*p* Value
MAP	−6 (95% CI −13.39–1.39)	0.11	−2.2 (95% CI −11.25–6.85)	0.63	+2.2 (95% CI −6.85–11.25)	0.63
CVP	+1.6 (95% CI −0.06–3.26)	0.06	+1.6 (95% CI −0.43–3.63)	0.12	+0.2 (95% CI −1.83–2.22)	0.85
CO	−0.09 (95% CI −0.28–0.96)	0.33	−0.04 (95% CI −0.27–0.19)	0.72	−0.06 (95% CI −0.29–0.17)	0.62
CPP	−6.8 (95% CI −14.51–0.91)	0.08	−4.2 (95% CI −13.64–5.24)	0.38	+1.6 (95% CI −7.84–11.04)	0.74

Abbreviations: MAP, mean arterial pressure (mmHg); CO, cardiac output (L/min); CPP, cerebral perfusion pressure (mmHg); CVP, central venous pressure (mmHg). Opening 1: first maximum opening pressure step (PIP 30; PEEP 15). Opening 2: second maximum opening pressure step (PIP 30; PEEP 15).

## Data Availability

The data that support the findings of this study are available from the corresponding author upon reasonable request.

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
