# Peer review of "Impact of Stepwise Recruitment Maneuvers on Cerebral Hemodynamics: Experimental Study in Neonatal Model"

_jpm, 2023, doi:10.3390/jpm13081184_

Round 1
Reviewer 1 Report
Summary: This study aimed to evaluate the impact of stepwise lung recruitment maneuvres (LRM) and individualised positive end expiratory pressure on cerebral hemodynamics in an animal model. The authors found that LRM were safely performed without any interruptions. There was systemic hemodynamic stability with no changes in intracranial pressure but this was associated with an increase in regional saturation oxygen saturations levels. The authors concluded that LRM is a safe tool to avoid atelectasis.
Abstract:
1. Please expand the abbreviations on first use. It may not be clear to the reader what these are.
2. Are these term equivalent piglets – it may be useful to mention this in the methods section
Introduction:
1. Expand all abbreviations on first use
2. Line 44 – VILI – the acronym is not expanded and the authors describe what it is but not the acronym. Either expand or remove it
3. What is RM? Please be consistent in the use of abbreviations
4. Line 58 – please re-phrase this sentence as it is unclear what the authors are trying to convey here
Material and Methods:
1. Line 72 – what were the animals examined by the veterinarian for?
2. Were the piglets term equivalent? Please comment
3. Why was the regional saturation monitor place on the left frontal level. Why not the right side and why not bilaterally. Please justify the reason for the selection so that the reader have a good understanding of the methodology
4. Why were the piglets left supine and not prone. Please justify/ provide reason for this.
5. Why were cuffed ETT not used?
6. How was ‘sealing’ the ETT to the trachea and guaranteeing the absence of leaks achieved? Would the narrowing that results from this practice influence the flow of gas from the ETT and secondly cause air flow disturbance within the trachea. Please discuss these in the discussion.
7. The Flow-I 4.3 anaesthesia system that was used for LRM; what is the range of measurements that this can be use for. Are there any studies examining the validity of this tool in LRM? How sure can the authors be that the intended pressures during LRM were achieved using this tool?
8. What was used to ensure alveolar collapse occurred and what was used to ensure that alveolar recruitment occurred with LRM
9. Line 144 – why were these time points selected. Please discuss.
Results:
1. Expand Cst. It would be really nice if the authors could list all the abbreviations used at the start of the study so that the reader have a clear understanding.
2. What is MR? Line 171
3. Table 2 – it is unclear what the authors mean by ‘opening 1’ and ‘opening 2’. In the LRM process, there are at least 3 steps of increment in PEEP. Have these been accounted for in the results?
4. What was the impact of central venous pressure on increasing PEEP?
5. All the graphs need to be bigger as they are difficult to follow
Discussion:
1. It may be useful to discuss around the impact of sevoflurane use on the physiological parameters that were examined in the study.
2. Could there be a chance that the true effects of the interventions may be masked by a small sample size? Authors need to discuss this and may consider changing the conclusion to account for this. Larger trials may be needed to examine the true relationship between these parameters. Please discuss
3. Are there other animal studies examining these relationships?
Conclusion:
1. Please modify accordingly.
The grammer can be improved
Author Response
Dear Reviewer 1:
First of all, we would like to thank you for your valuable time and effort spent reviewing our manuscript. Your considerations and suggestions have been of great help in significantly improving the quality and clarity of the paper.
I deeply appreciate your commitment and your observations have been extremely helpful in identifying and correcting areas of improvement, thus strengthening the coherence and rigor of the research submitted. Your expertise in this area has been invaluable in enriching the contribution of this study to our discipline.
We proceed to answer your questions, hoping to cover your expectations. However, in case that you want further considerations, please let us know. We have also highlighted the changes in green to make it easier to find the modifications and improvements we have introduced.
ABSTRACT
Q1. Please expand the abbreviations on first use. It may not be clear to the reader what these are.
Thank you very much for your appreciation. We have corrected in the text as you suggested.
Q2. Are these term equivalent piglets – it may be useful to mention this in the methods section???
Thank you for your appreciation. We have removed the term “piglets” from the title because it doesn’t provide additional information. We have clarified its meaning in the method section. The revised title will be in neonatal model
We have changed the term “piglets” to newborn pigs. Regarding newborn pigs are less than 72 hours old, and all of them exhibited a similar size with a weight around 2.56+/0 0,18 kg. If you are referring to the duration of animal preparation, all subjects underwent the same procedures in the same sequential order, with a duration of 35 +/- 3 minutes for all preceding procedures leading up to the LRM.
INTRODUCTION:
Q1. Expand all abbreviations on first use
Thank you very much for your suggestion. We have corrected in the text as you suggested.
Q2. Line 44 – VILI – the acronym is not expanded and the authors describe what it is but not the acronym. Either expand or remove it
Thank you very much for your appreciation. We have corrected in the text as you suggested.
Q3. What is RM? Please be consistent in the use of abbreviations.
We have made a transcription error and apologize for it. We referred to LRM (lung recruitment maneuvers). We have changed it in the text, sorry for that.
Q4. Line 58 – please re-phrase this sentence as it is unclear what the authors are trying to convey here???
Thank you very much for your comment. We have made the changes according to your suggestions, hoping that we have met your expectations. Please let us know if there is anything else we can improve.
“Evidence supports the safety of employing recruitment maneuvers in the neonatal population. LRM have been found to have no deleterious effects on systemic haemodynamics (including arterial blood pressure, central venous pressure and cardiac output), as well as on respiratory parameters (airtrapping and pneumothorax). However, the impact of these maneuvers on cerebral perfusion in neonatal brain models remains unknown.”
MATERIAL AND METHODS:
Q1. Line 72 – what were the animals examined by the veterinarian for?
Thank you very much for your comment. In all animal research studies, it is mandatory to have an external accredited veterinary doctor conduct a clinical examination of the study animals to ensure that they are in a good baseline health condition before starting the study. Occasionally, especially in neonatal pigs, there may be instances where an animal is not in perfect condition.
Q2. Were the piglets term equivalent? Please comment
Thank you very much for your comment.
We have changed the term piglets to newborn pigs. These are pigs that are neonates until weaning age, which is usually three weeks", in our study, we used newborn pigs less than 72 hours old, and all of them were of similar size with a weight of about 2.56+/-0 0.18 kg.
But we can maintain the term piglet or just plain pig. Let us know what you think is best.
Q3. Why was the regional saturation monitor place on the left frontal level? Why not the right side and why not bilaterally. Please justify the reason for the selection so that the reader has a good understanding of the methodology
We really appreciate your question. The reason for deciding to monitor the left side was purely technical. In a neonatal pig, the head circumference is only 6-8 cm, which makes it physically impossible to simultaneously monitor ICP, bilateral brain oximetry, and bilateral BIS (bispectral index). Therefore, a choice must be made to prioritize oximetry on one side instead of using bilateral monitoring (even with specific neonatal sensors). We utilize the opposite side to place the ICP catheter and secure it in place. Given that the piglets were neonatal and healthy, the rSO2 should be similar in both hemispheres. However, to avoid bias in placement, we made a systematic decision to monitor the left hemisphere in all animals equally.
Q4. Why were the piglets left supine and not prone. Please justify/ provide reason for this.
We really appreciate your comments. The decision to position the piglets in a supine position rather than a prone position is based on two fundamental reasons. Firstly, with the intracranial pressure catheter placed at the frontal level, the optimal way to control it was by keeping the patient in a supine position avoiding movements or displacements of the ICP catheter. Additionally, we aimed to replicate the experiments as closely as possible to real-life scenarios, and in both the operating room and neonatal intensive care units, the majority of our patients are in a supine position.
Q5. Why were cuffed ETT not used?
We really appreciate your question. Based in our previous experimental studies and the general practice in animal research, if you need that is not possible to have leaks in piglets is not enough to use cuffed ETT because the tracheal tissue is very elastics and when high inspiratory pressures are used cuffed ETT could leak despite being inflated. We choose to use an uncuffed ETT and sealing it to the trachea so we ensure the total absence of leaks that can make the results less precise and exact.
González-Pacheco N, Sánchez-Luna M, Arribas-Sanchez C, Santos-González M, Orden-Quinto C, Tendillo-Cortijo F. DCO2/PaCO2 correlation on high-frequency oscillatory ventilation combined with volume guarantee using increasing frequencies in an animal model. Eur J Pediatr 2020;179(3):499-506. doi: 10.1007/s00431-019-03503-8.
González-Pacheco N, Sánchez-Luna M, Chimenti-Camacho P, Santos-González M, Palau-Concejo P, Tendillo-Cortijo FJ. Use of very low tidal volumes during high-frequency ventilation reduces ventilator lung injury. J Perinatol 2019;39(5):730-736. doi:10.1038/s41372-019-0338-5.
Serrano Zueras C, Guilló Moreno V, Santos González M, Gómez Nieto FJ, Hedenstierna G, García Fernández J. Safety and efficacy evaluation of the automatic stepwise recruitment maneuver in the neonatal population: An in vivo interventional study. Can anesthesiologists safely perform automatic lung recruitment maneuvers in neonates? Pediatr Anesth. 2021 Sep;31(9):1003–10
Q6. How was ‘sealing’ the ETT to the trachea and guaranteeing the absence of leaks achieved? Would the narrowing that results from this practice influence the flow of gas from the ETT and secondly cause air flow disturbance within the trachea. Please discuss these in the discussion.
Thank you very much for your comment. To guarantee the absence of leaks during the recruitment maneuver, we opted the to secure the ETT to the trachea. Surgical dissection was performed in planes, exposing the trachea. Once exposed, a tracheal tube was placed and sealed with a ribbon wrapped around the trachea to prevent any leak. The sealed with a ribbon wrapped prevents any injury or occlusion to the trachea. We confirmed that the ligature effectively sealed the endotracheal tube (ETT) without causing any narrowing of its diameter, based on the absence of any observable changes in pressures before and after the trachea was ligated. Absence of leaks was determined through the analysis of the flow-volume.
González-Pacheco N, Sánchez-Luna M, Arribas-Sanchez C, Santos-González M, Orden-Quinto C, Tendillo-Cortijo F. DCO2/PaCO2 correlation on high-frequency oscillatory ventilation combined with volume guarantee using increasing frequencies in an animal model. Eur J Pediatr 2020;179(3):499-506. doi: 10.1007/s00431-019-03503-8.
González-Pacheco N, Sánchez-Luna M, Chimenti-Camacho P, Santos-González M, Palau-Concejo P, Tendillo-Cortijo FJ. Use of very low tidal volumes during high-frequency ventilation reduces ventilator lung injury. J Perinatol 2019;39(5):730-736. doi:10.1038/s41372-019-0338-5.
Serrano Zueras C, Guilló Moreno V, Santos González M, Gómez Nieto FJ, Hedenstierna G, García Fernández J. Safety and efficacy evaluation of the automatic stepwise recruitment maneuver in the neonatal population: An in vivo interventional study. Can anesthesiologists safely perform automatic lung recruitment maneuvers in neonates? Pediatr Anesth. 2021 Sep;31(9):1003–10
Q7. The Flow-I 4.3 anaesthesia system that was used for LRM; what is the range of measurements that this can be used for. Are there any studies examining the validity of this tool in LRM? How sure can the authors be that the intended pressures during LRM were achieved using this tool?
Thank you for your concern about the validation of the Flow-i 4.3 software for LRM. As is customary in the industry, when software is launched on the market, it is subjected to rigorous validation and security testing before it is released to the market.
Specifically, for Flow-i 4.3 software, the manufacturer has carried out an extensive validation process, covering functionality, performance and security testing. The aim of this testing is to ensure that the software meets the established standards and requirements before being used in clinical environments.
There are studies demonstrating the effectiveness of these maneuvers (with improvements in oxygenation and compliance) and the safety of these maneuvers, without deleterious effects on systemic hemodynamics or respiratory complications.
Serrano Zueras C, Guilló Moreno V, Santos González M, Gómez Nieto FJ, Hedenstierna G, García Fernández J. Safety and efficacy evaluation of the automatic stepwise recruitment maneuver in the neonatal population: An in vivo interventional study. Can anesthesiologists safely perform automatic lung recruitment maneuvers in neonates? Pediatr Anesth. 2021 Sep;31(9):1003–10.
Guilló-Moreno V, Gutiérrez-Martínez A, Serrano-Zueras C, Santos-González M, Romero-Berrocal A, García-Fernández J. Shortened Automatic Lung Recruitment Maneuvers in an In Vivo Model of Neonatal ARDS. Respir Care. 2023 May;68(5):628–37.
The aim of our study is to assess whether there were also no deleterious effects on the central nervous system.
Q8. What was used to ensure alveolar collapse occurred and what was used to ensure that alveolar recruitment occurred with LRM
Thank you for your observation. “Atelectasis and poorly ventilated lung areas appear during general anesthesia in adults as well as in children”. “It is known that atelectasis occurs in the most dependent parts of the lung of 90% of patients who are anesthetized and plays an important role in gas exchange abnormalities and reduced static compliance.”
During general anesthesia after the induction, the diaphragm relaxes and moves cephalically, leading to a greater increase in pleural pressure in dependent lung regions and adjacent lung tissue can be compressed.
Lundquist H, Hedenstierna G, Strandberg A, Tokics L, Brismar B: CT- assessment of dependent lung densities in man during general anaesthesia. Acta Radiol 1995; 36:626 –32
Brismar B, Hedenstierna G, Lundquist H, Strandberg A, Svensson L, Tokics L: Pulmonary densities during anesthesia with muscular relaxation: A proposal of atelectasis. Anesthesiology 1985; 62:422– 8, doi:10.1097/00000542-198504000-00009
Healthy neonatal model is the perfect model for collapse. Neonates collapse their lung in 100% of cases after induction of anesthesia because the Functional Residual Capacity is lower than the End-Expiratory Lung Volume.
Tusman, G.; Böhm, S.H.; Tempra, A.; Melkun, F.; García, E.; Turchetto, E.; Mulder, P.G.H.; Lachmann, B. Effects of Recruitment Maneuver on Atelectasis in Anesthetized Children. Anesthesiology 2003, 98, 14 22, doi:10.1097/00000542-200301000-00006.
Neumann RP, Von Ungern- Sternberg BS. The neonatal lungphysiology and ventilation. Paediatr Anaesth. 2014;24:10-21.
Garcia- Fernandez J, Castro L, Belda J. Ventilating newborn and child. Curr Anaesth Crit Care. 2010;21:262-268.
Harris T. Physiological principles. In: Goldsmith JP, Karotkined EH, editors. Assisted ventilation of the neonate. Philadelphia: WB Saunders; 1988:P22-P69.
Additionally, we measured the dynamic compliance (Cdyn) before and after the procedure and compared it with the theoretically expected value based on ideal body weight.
Suarez-Sipmann F, Böhm SH, Tusman G, Pesch T, Thamm O, Reissmann H, et al. Use of dynamic compliance for open lung positive end-expiratory pressure titration in an experimental study. Crit Care Med 2007;35(1):214-221.
We also observed an improvement in Cstat after LRM, indicating the presence of collapsed lung zones.
Q9. Line 144 – why were these time points selected. Please discuss.
Thank you for your suggestion. We chose those points for the following reasons:
- Start of the maneuver: We selected this point to assess the baseline data of the piglets after inducing general anesthesia and intubation, which may lead to partial collapse of the lungs. By examining the data at this stage, we can understand the initial conditions and any changes induced by the anesthesia and intubation process.
- During the maximum opening pressure step: This point was chosen to evaluate the effects of increased intrathoracic pressure on systemic hemodynamics and cerebral parameters. It allows us to examine the response and potential adverse effects of the maneuver at the maximum pressure level.
- 10 minutes after the end of LRM: We selected this point to determine the values once the maneuver is completed and the alveoli are maximally opened. By examining the data at this stage, we can assess the immediate effects of the maneuver and evaluate any changes in the lung, systemic haemodynamics and cerebral parameters.
RESULTS:
Q1. Expand Cst. It would be really nice if the authors could list all the abbreviations used at the start of the study so that the reader have a clear understanding.
Thank you very much for your suggestion. In the format of this manuscript and this scientific journal, we are unsure of the appropriate section to incorporate a list of the abbreviations used. Here, we provide the complete list of abbreviations used in the text:
LRM: Lung Recruitment Maneuver
PEEP: positive end-expiratory pressure
PIP: peak inspiratory pressure
MAP: Mean Arterial Pressure
CVP: Central Venous Pressure
CO: Cardiac Output
ICP: Intracranial Pressure
rSO2: Cerebral Oxygen Saturation
Cstat: Static Compliance
Cdyn: Dynamic Compliance
DP: Driving pressure
TV: Tidal Volume
SpO2: Peripheral O2 Saturation
VCV: Volume-controlled Ventilation
RR: Respiratory rate
EtCO2: End tidal CO2
Pmean: Mean Airway Pressure
HR: Heart Rate
BP: Blood pressure
SBP: Systolic blood Pressure
DBP: Diastolic Blood Pressure
We recognize the importance of defining all abbreviations used in the text to facilitate reader comprehension.
Q2. What is MR? Line 171
Thank you for your comment. We have made a transcription error and apologize for it. We referred to LRM (lung recruitment maneuvers).
Q3. Table 2 – it is unclear what the authors mean by ‘opening 1’ and ‘opening 2’. In the LRM process, there are at least 3 steps of increment in PEEP. Have these been accounted for in the results?
We really appreciate your comment. We have proceeded to describe the meaning of Opening 1 and Opening 2 which are the lung recruitment maneuvers steps with maximum inspiratory pressure (30 cmH2O) and maximum PEEP (15 cmH2O).
Although stepwise LRM involves a gradual increase in pressure in multiple steps, we have chosen to collect data at the points of maximum pressure (both inspiratory and PEEP) as they represent the instances where there may be significant respiratory, hemodynamic, and cerebral impact.
Furthermore, lung expansion does not occur during the preceding stages but rather at the moment of maximal pressure step. This is because the sole pressure that effectively opens the collapsed lung regions is the maximum transpulmonary pressure, as opposed to the PEEP. Consequently, the complete opening process takes place when the critical opening pressure is surpassed, which only happens when the transpulmonary pressure exceeds 25 cmH2O. No substantial opening process occurs during the earlier stages.
Gattinoni, L., Caironi, P., Cressoni, M., Chiumello, D., Ranieri, V.M., Quintel, M., et al (2006). Lung recruitment in patients with the acute respiratory distress syndrome. New England Journal of Medicine, 354(17), 1775-1786(1).
Talmor, D., Sarge, T., Malhotra, A., O'Donnell, C.R., Ritz, R., Lisbon, A., & Loring, S.H. (2008). Mechanical ventilation guided by esophageal pressure in acute lung injury. New England Journal of Medicine, 359(20), 2095-2104.
Tusman G, Bohm S, Tempra A, et al Effects of recruitment maneuver on atelectasis in anesthetized children. Anesthesiology 2003; 98:14-22
Hess DR. Recruitment Maneuvers and PEEP Titration. Respir Care. 2015 Nov 1;60(11):1688–704.
Q4. What was the impact of central venous pressure on increasing PEEP?
Thank you for your question.
Numerous studies have examined the impact of positive end-expiratory pressure (PEEP) on central venous pressure (CVP) in various clinical scenarios. These studies have demonstrated an increase in CVP with increasing PEEP levels.
Sand L, Rizell M, Houltz E, Karlsen K, Wiklund J, Odenstedt Hergès H, et al. Effect of patient position and PEEP on hepatic, portal and central venous pressures during liver resection: Patient position during liver resection. Acta Anaesthesiol Scand. 2011 Oct;55(9):1106–12.
Jha L, Lata S, Jha AK, Prasad SK. Effect of positive end-expiratory pressure on central venous pressure in the closed and open thorax. Physiol Meas. 2022 Aug 31;43(8):085006.
While there is a study indicating that the increase in CVP due to PEEP does not have a detrimental effect on the elevation of intracranial pressure (ICP), it has been observed that dangerous increases in ICP depend on ICP gap (baseline ICP - baseline CVP).
Li HP, Lin YN, Cheng ZH, Qu W, Zhang L, Li QY. Intracranial-to-central venous pressure gap predicts the responsiveness of intracranial pressure to PEEP in patients with traumatic brain injury: a prospective cohort study. BMC Neurol. 2020 Dec;20(1):234
However, in our study, we did not observe any changes in CVP during the LRM, possibly due to the limited duration (less than a minute). There were also no variations in ICP during the LRM.
Q5. All the graphs need to be bigger as they are difficult to follow
Thank you very much for your appreciation. We have corrected in the manuscript as you suggested.
DISCUSSION:
Q1. It may be useful to discuss around the impact of sevoflurane use on the physiological parameters that were examined in the study.
Thank you for your suggestion. We have chosen sevoflurane as the anesthetic agent for our study. This inhalational anesthetic has been the subject of numerous studies investigating its impact on systemic hemodynamics. Our decision was based on its ability to maintain adequate hemodynamic stability with minimal effects on blood pressure and systemic resistance. Additionally, sevoflurane has demonstrated the ability to maintain stable cardiac output during anesthesia, making it a safe choice.
Regarding its relationship with ICP and rSO2, several studies have examined these aspects in patients undergoing intracranial surgery. Sevoflurane has been found to exert neuroprotective effects, as it can reduce ICP and improve rSO2 compared to other anesthetic agents. Furthermore, studies have shown that sevoflurane decreases cerebral metabolic rate of oxygen (CMRO2) while maintaining a proportional reduction in cerebral blood flow (CBF). This preserved ratio of CBF to CMRO2 supports its utility in neuroanesthesia, indicating its potential to preserve cerebral perfusion and mitigate neuronal damage.
Juhász, M.; Molnár, L.; Fülesdi, B.; Végh, T.; Páll, D.; Molnár, C. Effect of Sevoflurane on Systemic and Cerebral Circulation, Cerebral Autoregulation and CO2 Reactivity. BMC Anesthesiol. 2019, 19, 109, doi:10.1186/s12871-019-0784-9.
Valencia, L.; Rodríguez-Pérez, A.; Kühlmorgen, B.; Santana, R.Y. Does Sevoflurane Preserve Regional Cerebral Oxygen Saturation Measured by Near-Infrared Spectroscopy Better than Propofol? Ann. Fr. Anesth. Réanimation 2014, 33, e59–e65, doi:10.1016/j.annfar.2013.12.020.
The dose of sevoflurane that we used was the minimum required to induce unconsciousness in the animal. It is worth noting that the minimum alveolar concentration (MAC) in neonates is higher than in adults (see Materials and Methods).
Lerman, J.; Oyston, J.P.; Gallagher, T.M.; Miyasaka, K.; Volgyesi, G.A.; Burrows, F.A. The Minimum Alveolar Concentration (MAC) and Hemodynamic Effects of Halothane, Isoflurane, and Sevoflurane in Newborn Swine. Anesthesiology 1990, 73, 717–721, doi:10.1097/00000542-199010000-00018.
Q2. Could there be a chance that the true effects of the interventions may be masked by a small sample size? Authors need to discuss this and may consider changing the conclusion to account for this. Larger trials may be needed to examine the true relationship between these parameters. Please discuss
As our study is an experimental animal design is limited to a small sample size due to the ethical limitations of the ARRIVE guidelines. Although we calculated the sample size to be sufficient to obtain statistically significant results (included in text section statistical analysis), it would be interesting to conduct clinical studies increasing the number of patients to verify whether our results are confirmed in studies with larger sample sizes.
However, even though it would be ideal to have confirmatory clinical trials in humans that would validate these findings, it is ethically impossible to conduct prospective studies on healthy neonates with such an invasive level of monitoring.
Q3. Are there other animal studies examining these relationships?
In our discussion, we have compared our work with similar studies.
Chen, H.; Zhou, X.-F.; Zhou, D.-W.; Zhou, J.-X.; Yu, R.-G. Effect of Increased Positive End-Expiratory Pressure on Intracranial Pressure and Cerebral Oxygenation: Impact of Respiratory Mechanics and Hypovolemia. BMC Neurosci. 2021, 22, 72, doi:10.1186/s12868-021-00674-9.
Chen, H.; Zhou, J.; Lin, Y.-Q.; Zhou, J.-X.; Yu, R.-G. Intracranial Pressure Responsiveness to Positive End-Expiratory Pressure in Different Respiratory Mechanics: A Preliminary Experimental Study in Pigs. BMC Neurol. 2018, 18, 183, doi:10.1186/s12883-018-1191-4.
The reason for conducting our experimental study was the lack of existing research on the impact of automatic recruitment maneuvers on cerebral hemodynamics (ICP and SrO2) in healthy neonates. This field is underexplored due to the technical difficulties involved in studying the neonatal model and the impossibility of conducting clinical studies in human neonates.
We have tried to answer the questions that you and other reviewers have made. We submit you again our manuscript with the modifications so that you can reconsider it for publication in the Journal. If you have any further suggestions or comments on the changes introduced, we would be delighted to receive your additional comments. We would greatly value any new perspective or approach that you consider relevant to further refine the work.
Once you have positively assessed the corrections made, we will send the manuscript to MDPI's linguistic editing service.
Once again, we sincerely thank you for your time and invaluable contribution to our research.
Best regards
Reviewer 2 Report
The purpose of this article was to evaluate the impact of stepwise LRM and individualized PEEP on cerebral hemodynamics in an experimental neonatal model.
This an interesting work, though the sample is quite small and it refers to animals and not to humans.
There are some comments for the authors:
1. The text needs some editing regarding spelling, especially the part with the results.
2. In line 44 ‘’ since they provide…’’ needs correction from plural to singular.
3. In line 169 “there are improvements…”, please correct In the right tense.
4. In line 163 please correct spelling.
5. Table 2 needs editing. ‘’Opening 1 and opening 2’’ are not explained to what refer.
6. In line 211 the abbreviation ‘’GC’’ is not analyzed.
7. Figure 4 does not include GC.
the text in general needs some editing regarding spelling
Author Response
Dear Reviewer 2:
First of all, we would like to thank you for your valuable time and effort spent reviewing our manuscript. Your considerations and suggestions have been of great help in significantly improving the quality and clarity of the paper.
I deeply appreciate your commitment and your observations have been extremely helpful in identifying and correcting areas of improvement, thus strengthening the coherence and rigor of the research submitted. Your expertise in this area has been invaluable in enriching the contribution of this study to our discipline.
We proceed to answer your questions, hoping to cover your expectations. However, in case that you want further considerations, please let us know. We have also highlighted the changes in yellow to make it easier to find the modifications and improvements we have introduced.
Thank you for your appreciation about the sample. We calculated the number of piglets in order to ensure the accuracy of the results and reduced to the least number of animals according of ARRIVE guidelines.
The reason why the study was conducted in piglets instead of humans is because invasive monitoring (intracranial pressure catheter, arterial pressure catheter placement and central venous line placement) is contraindicated in healthy neonates. We thought it would be interesting to study the effect of lung recruitment maneuvers on cerebral and systemic haemodynamics, and although we would have liked to carry it out in humans, ethically it should not be done.
Q1: The text needs some editing regarding spelling, especially the part with the results.
Thank you very much for your comment. Once we have sent you the new version of the manuscript and you have positively assessed the corrections made, we will send the manuscript to MDPI's linguistic editing service.
Q2: In line 44 ‘’ since they provide...’’ needs correction from plural to singular.
Thank you very much for your appreciation. We have corrected in the text as you suggested.
Q3. In line 169 “there are improvements...”, please correct in the right tense.
Thank you very much for your appreciation. We have corrected in the text as you suggested.
Q4. In line 163 please correct spelling.
Thank you very much for your appreciation. We have corrected in the text as you suggested.
Q5. Table 2 needs editing. ‘’Opening 1 and opening 2’’ are not explained to what refer.
Thank you very much for your appreciation. We have proceeded to describe the meaning of Opening 1 and Opening 2 which are the lung recruitment maneuvers steps with maximum inspiratory pressure (30 cmH2O) and maximum PEEP (15 cmH2O).
Q6. In line 211 the abbreviation ‘’GC’’ is not analyzed.
We have made a transcription error and apologize for it. GC stands for cardiac output in Spanish. We have corrected the mistake by translating it into CO
Q7. Figure 4 does not include GC.
We are embarrassed and apologize for the mistake (Q6). We have corrected it already and modified GC by CO. Figure 4 includes the linear prediction of ICP (intracranial pressure) according to the CO
We have tried to answer the questions that you and other reviewers have made. We submit you again our manuscript with the modifications so that you can reconsider it for publication in the Journal. If you have any further suggestions or comments on the changes introduced, we would be delighted to receive your additional comments. We would greatly value any new perspective or approach that you consider relevant to further refine the work.
Once again, we sincerely thank you for your time and invaluable contribution to my research.
Best regards
Round 2
Reviewer 1 Report
Thank you very much for addressing all the queries and thank you for your contribution. There are minor corrections required.
Thank you very much for addressing all the queries and thank you for your contribution. There are minor corrections required.